# Does a novel diagnostic pathway including blood-based risk prediction and MRI-targeted biopsies outperform prostate cancer screening using prostate-specific antigen and systematic prostate biopsies? - protocol of the randomised study STHLM3MRI

Tobias Nordström,[1,2] Fredrik Jäderling,[3,4] Stefan Carlsson,[5] Markus Aly,[1,3,5] Henrik Grönberg,[1] Martin Eklund[1]

For numbered affiliations see end of article.

**Correspondence to**
Tobias Nordström;
tobias.nordstrom@ki.se

## ABSTRACT

**Introduction** Prostate cancer is a leading cause of cancer death among men in the Western world. Early detection of prostate cancer has been shown to decrease mortality, but has limitations with low specificity leading to unnecessary biopsies and overdiagnosis of low-risk cancers. The STHLM3 trial has paved way for improved specificity in early detection of prostate cancer using the blood-based STHLM3 test for identifying men at increased risk of harbouring significant prostate cancer. Targeted prostate biopsies based on MRI images have shown non-inferior sensitivity to detect significant prostate cancer and decrease the number of biopsies and non-significant cancers among men referred for prostate biopsy in clinical practice. The strategy of the STHLM3-MRI project is to study an improved diagnostic pathway including an improved blood-based test for identification of men with increased risk of prostate cancer and use of MRI to select men for diagnostic workup with targeted prostate biopsies.

**Methods** This study compares prostate cancer detection using prostate-specific antigen (PSA) and systematic biopsies to the improved pathway for prostate cancer detection using the STHLM3 test and targeted biopsies in a screening context. The study will recruit 10 000 participants during 1 June 2018 to 1 June 2020 combining a paired and randomised design. Participants are grouped by PSA and Stockholm3 test level. Men with Stockholm3 ≥11% or PSA ≥3 ng/mL are randomised to systematic or MRI-targeted biopsies. This protocol follows SPIRIT guidelines. Endpoints include the number of detected prostate cancers, number of performed biopsy procedures and number of performed MRIs. Additional aims include to assess the health economic consequences and development of automated image-analysis.

**Ethics and dissemination** The study is approved by the regional ethical review board in Stockholm (2017-1280/31). The study findings will be published in peer-review journals. Findings will also be disseminated by conference/departmental presentations and by media.

## Strengths and limitations of this study

► This is the first randomised study to examine the role of improved blood-based risk stratification used in sequence with MRI and targeted prostate biopsies in a screening-by-invitation context.

► The study examines the performance of the Stockholm3 test used together with MRI/Fusion technique compared with traditional prostate-specific antigen screening and will provide important data also on the performance of the Stockholm3 test or MRI/Fusion when used as standalone strategies.

► The study is performed at three study sites and uses centralised radiology and pathology.

► The study is limited to a Swedish screening population, the use of the Stockholm3 test as blood-based risk prediction test and the technology used for MRI-targeted biopsies.

**Trial registration number** NCT03377881; Pre-results.

## INTRODUCTION

### Public health significance of prostate cancer

Prostate cancer is the most common cancer and the leading cause of cancer death among men in Sweden. In year 2011, over 10 000 men were diagnosed with prostate cancer and more than 2500 died due to the disease, approximately 20% of these in the Stockholm region. Prostate cancer incidence rates in Sweden are now comparable to rates in countries that had an early introduction of prostate-specific antigen (PSA) testing, while prostate cancer mortality rates are higher than in most other countries.[1] With over

90 000 prevalent cases, the health burden and the costs on the healthcare system are substantial. While a number of risk factors have been proposed for prevention of prostate cancer, including diet and occupational exposures, the only factors conclusively shown to increase risk of the disease are age, ethnicity and family history. Given the high prevalence of the cancer and limited opportunities for primary prevention, improved detection would reduce both procedure-related harm to men and economical cost in the healthcare system.

### Early detection and treatment of prostate cancer: benefits and harms

The PSA test was first used to monitor disease progression in prostate cancer patients. The PSA test was taken up as a de facto screening test for prostate cancer in many countries, leading to a rapid rise in prostate cancer incidence. The test characteristics for the PSA test in detecting prostate cancer are comparable to those for mammography for breast cancer screening, with a sensitivity of 72% and a specificity of 30%–35% at a test threshold of 4 ng/mL.[2] However, a lower threshold of 3 ng/mL adopted in Sweden recently has led to increased sensitivity at the expense of reduced specificity. Recent analyses of PSA testing in the Stockholm area confirms these results showing that 46%, 68% and 77% of men 50–59, 60–69 and 70–79 years, respectively, have had at least one PSA test during a 9 years period.[3]

Recent results from the large European Randomized Study of Screening for Prostate Cancer (ERSPC) including over 180 000 men provide increasing evidence that PSA screening has led to reduced mortality.[4] This report showed that PSA screening without digital rectal examination (DRE) was associated with a 21% relative reduction in the death rate from prostate cancer at a median follow-up of 11 years, with an absolute reduction of about 7 prostate cancer deaths per 10 000 men screened. Estimations from the ERSPC trial (men aged 55–69 years) show that 1048 men would need to be offered screening and an additional 37 would need to be managed to prevent one prostate cancer death during a 10-year period, leading to a significant overtreatment of indolent disease. The effectiveness of PSA testing was more marked at the Göteborg site of the ERSPC trial, with a risk reduction of 44% over 14 years in men aged 50-64 years.[5] This effect size is larger than that observed for mammographic screening for breast cancer and faecal occult blood testing for colorectal cancer.

However, using traditional systematic biopsies for diagnosis, approximately half of diagnosed cancers are low-risk tumours using the same main cut-off for biopsy as the ERSPC trial (PSA=3 ng/mL).[6 7] It has been shown that men with low-risk tumours treated without curative intent have the same survival as men in the background population,[8] illustrating the large proportion of overdiagnosed cancers.[9]

The STHLM3 study has shown one way to improve identification of men at increased risk of significant prostate cancer. Using the STHLM3 test, 32% of the prostate biopsies may be saved while not decreasing the sensitivity to high-grade disease (defined as Gleason Score ≥7) and simultaneously decreasing the number of low-grade tumours (Gleason Score ≤6) by 17%, thus decreasing overdiagnosis.[7]

### Traditional evaluation of men with increased risk of prostate cancer

Men at increased risk of prostate cancer—commonly estimated using PSA and palpatory findings—are traditionally assessed using systematic prostate biopsies. The procedure is performed under local anaesthesia using antibiotic prophylaxis and includes 10–12 cores taken from predefined areas of the peripheral zone of the gland as visualised by endorectal ultrasound. While the biopsies systematically covers the prostatic gland rather than targeting a specific lesion, and non-lethal tumours are common, the risk of overdiagnosis (ie, detection of non-significant tumours) is high.[9] The risk of non-representative biopsy findings results in underestimation of tumour grade compared with subsequent prostatectomy in up to 40% of men undergoing surgery.[10] The risk of severe post-biopsy infection has increased to 1%–2% with increasing frequency of antibiotic resistance, further illustrating the need both to increase precision and decrease the number of performed biopsies.[11] Since screening using PSA and systematic prostate biopsies have been shown to decrease prostate cancer mortality, it is reasonable to use this strategy as comparator for novel diagnostic strategies.[4 5]

### Multiparametric magnetic resonance imaging for detection of prostate cancer

Multiparametric magnetic resonance imaging (mpMRI) incorporating anatomical and functional imaging has now been validated as a means of detecting and characterising prostate tumours and can aid in risk stratification and treatment selection. The European Society of Urogenital Radiology in 2012 established the Prostate Imaging Reporting and Data System (PI-RADS) guidelines aimed at standardising the acquisition, interpretation and reporting of prostate mpMRI. Consensus on an updated version (PI-RADS V.2) has recently been published, outlining aspects of both interpretation and the technical execution.[12–14] Use of the revised PI-RADS provides moderately reproducible MR imaging scores for detection of clinically relevant disease.[15] Using mpMRI to triage men might allow 27% of patients to avoid a primary biopsy and enable a decrease in the detection of clinically insignificant cancers. If subsequent transrectal ultrasound-guided (TRUS) biopsies were directed by mpMRI findings, up to 18% more cases of clinically significant cancer might be detected compared with the standard pathway of TRUS biopsy for all.[16]

In summary, PI-RADS recommends to use 3T or 1.5T machines, including T2-weighted and T1-weighted sequences together with diffusion weighted images.

Currently, the added value of dynamic contrast is not firmly established regarding tumour detection. At this time, there is no consensus among experts concerning the potential benefits of the use of endorectal coils for cancer detection. It has been suggested that the prevalence of suspicious lesions on MRI in men with clinical suspicion of prostate cancer is approximately 60%.[17]

### Targeted prostate biopsies guided by fusion technology

Targeted biopsies of the prostate consist of imaging (MRI) detecting significant tumours and a biopsy procedure where biopsies are targeted to the tumour using various devices for guidance.[18] While traditional endorectal ultrasound poorly identifies tumours, direction of biopsy needles can be performed in various ways. Cognitive or soft fusion is based on skilled urologists/radiologists interpreting the MRI images and directing needles solely based on the ultrasound images. The disadvantages of cognitive fusion lie in the potential for human error when attempting to mentally fuse the MRI with TRUS while aiming for cancers that are often <1 cm in diameter and the inability to track the location of each biopsy site. Hard fusion enables proper fusion of MRI information on the ultrasound image, possibly increasing precision.

Despite methodological flaws, a number of studies have investigated the value of fusion biopsies, primarily using non-randomised designs and non-screening populations.[19] In 2018, Kasivisvanathan *et al* provided high-quality evidence for men referred for prostate biopsy and showed that MRI/target biopsies are non-inferior for detection of significant cancer and decreases the number of insignificant cancers and number of biopsies when compared with systematic biopsies.[20]

The proportion of men upgraded when comparing specimen from targeted biopsies and subsequent prostatectomy has been shown to be very low (<5%) when using targeted biopsies,[21] increasing the proportion of men where treatment decisions are based on valid risk estimations.

### Improving the diagnostic pathway for prostate cancer detection

The current diagnostic pathway for prostate cancer detection is characterised by several challenging hallmarks. First, testing with PSA is frequent also in men not benefitting from testing due to low PSA levels or high age.[3] Second, the currently used test for detection (PSA) lacks in specificity, resulting in frequent overdiagnosis.[22 23] Third, systematic biopsies show high frequencies of benign tests, overdiagnosis, upgrading at prostatectomy and risk of infectious complications.[7 24] Further, PSA testing increases with educational length, and men with high education are more likely to have a prostate biopsy after an increased PSA value. These differences may contribute to the worse prostate cancer outcomes observed among men with lower socioeconomic status.[25]

The STHLM3 test offers improved disease detection.[7] To further decrease overdetection, improve disease classification and spare men of test-related harm, prostate biopsy practices need to be improved. We hypothesise that an improved pathway for prostate cancer detection including a better blood-based screening test, improved selection to biopsy based on MRI findings and targeted biopsies guided by MRI/ultrasound fusion would dramatically decrease the number of biopsy procedures, overdiagnosis and improve treatment decisions.

## METHODS
### Hypotheses
#### Primary hypotheses

The following hypothesis is posed for men in screening-by-invitation context.

A diagnostic pathway using the Stockholm3 test to select men for further workup using MRI followed by targeted biopsies and systematic biopsies (S3M-MR-TBx/SBx) has non-inferior sensitivity for detecting clinically significant cancer (International Society of Urological Pathology (ISUP) grade group ≥2) and shows superior specificity (reduction in number of performed biopsy procedures and detected ISUP 1 tumours) compared with the diagnostic pathway using systematic biopsies in men with PSA ≥3 ng/mL (PSA-SBx).

#### Additional hypotheses

1. When compared with performing systematic biopsies for men with elevated risk of prostate cancer in prostate cancer screening, targeted prostate biopsies performed with MRI/Fusion technique with or without addition of systematic biopsies has non-inferior sensitivity for detecting clinically significant cancer (ISUP grade group ≥2) and reduces the number of performed biopsy procedures.
2. A diagnostic pathway using the Stockholm3 test to select men for further workup using MRI followed by ONLY targeted biopsies (S3M-MR-TBx) has non-inferior sensitivity for detecting clinically significant cancer (ISUP grade group ≥2) and reduces the number of performed biopsy procedures compared with a diagnostic pathway using systematic biopsies in men with PSA ≥3 ng/mL (PSA-SBx).
3. Adding prostate volume as parameter in the diagnostic pathway with Stockholm3 test and MRI/Fusion biopsies improves model precision.
4. A diagnostic pathway with Stockholm3 followed by MRI and targeted biopsies has non-inferior sensitivity for detecting clinically significant cancer (ISUP grade group ≥2) and reduces the number of MRI examinations and performed biopsies compared with a diagnostic pathway using PSA ≥3 ng/mL followed by MRI and targeted biopsies.
5. SBx in the MRI arm has superior sensitivity than SBx in the non-MRI arm (due to cognitive fusion).
6. Biopsy compliance is higher after biopsy is recommended based on MRI compared with recommended without MRI.

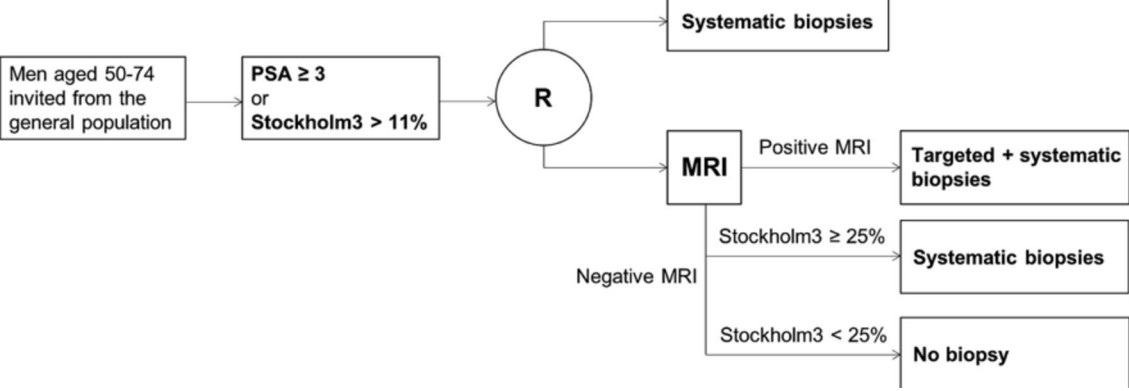

**Figure 1** Study design overview of STHLM3MRI Main Study.

7. A diagnostic pathway using the Stockholm3 test to select men for further workup using MRI and targeted biopsies (S3M+TBx) shows better health economy (positive incremental cost-effectivesness ratio (ICER)) compared with a diagnostic pathway using systematic biopsies in men with PSA ≥3 ng/mL (PSA +SBx).

## Aims

To compare a diagnostic pathway using the Stockholm3 test for selection of men to MRI and targeted biopsies (S3M+TBx) to a diagnostic pathway using systematic biopsies in men with PSA ≥3 ng/mL (PSA +SBx) with respect to number of diagnosed clinically significant cancer (ISUP grade group ≥2) and number of performed biopsies. Additional aims corresponding to Additional hypotheses 2–7 above will be assessed.

## Study design

STHLM3-MR phase II is a study combining a paired and a randomised design (figure 1). The study will follow the following outline: participants will be invited by mail. All participants will undergo a blood test, including PSA and the STHLM3 test. Men with an elevated PSA ≥3 ng/mL or PSA ≥1.5 ng/mL and S3M >11% will be randomised to either traditional prostate biopsies or MRI with targeted biopsies on MR lesions.

## Participants, interventions and outcomes
### Study setting

This is a screening-by-invitation study including one study administrative centre, two radiological sites and three urological sites where data will be collected.

### Participating urological centres

Department of Urology, Capio St Görans Hospital: dr Henrik Grönberg

Uroclinic, Sophiahemmet, Stockholm; dr Olof Jansson Odenplans läkarhus; dr Magnus Annerstedt

### Eligibility criteria
### Inclusion criteria
► Men aged 50–74 years without prior diagnosis of prostate cancer (ICD-9 C61).
► Permanent postal address in Stockholm.

► Not a previous participant in the Stockholm3 study (2012–2014).

### Exclusion criteria
► Severe illnesses such as metastatic cancers, severe cardiovascular disease or dementia.
► Contraindications for MRI, for example, pacemaker, magnetic cerebral clips, cochlear implants or severe claustrophobia.
► Men with a previous prostate biopsy the preceding 60 days before invitation.

### Randomisation

Randomisation is performed in the ratio 2:3 between control arm and experimental arm. Randomisation will be performed using stratification on disease risk (six strata). Disease risk is assessed using the Stockholm3 test. Tests are discordant if PSA is negative and Stockholm3 is positive or vice versa.

Four allocation lists (high/low risk vs discordant/concordant tests) have been created, specifying the sequence of study arm allocation (control arm, control arm, experimental arm, experimental arm, experimental arm). Participants are first allocated to corresponding list, and then allocated to study arm according to the order in which they participate. The allocation sequence is blinded from the study investigators and handled by the study database administrator (SDA, A Björklund).

In order to enhance resource usage, men are allocated to the study sites according to local availability of biopsy procedure slots.

### Interventions
### Blood sampling

Participating men undergo blood sampling with analysis of PSA and the Stockholm3 test at Karolinska University Laboratory.

For the main analysis, the Stockholm3 test includes clinical data as answered when consenting participation (previous biopsy, age, finasteride medication, relatives with prostate cancer); single nucleotide polymorphisms and measurements of protein levels (MSMB, MIC1, PSA, fPSA, hK2).[7] For secondary analyses, clinical information

on digital rectal examination (DRE) and prostate volume is included.

### Definition of experimental arm

Men randomised to the experimental arm undergo MRI. If suspicious lesions are found, the participant undergoes targeted biopsies using Fusion technology followed by systematic biopsies.

Men without lesions are excepted from further intervention and receives notification on recommendation for follow-up. Technology and process are described below.

Men with a Stockholm3 risk ≥25% and no suspicious lesion on MRI will be recommended to undergo systematic biopsies.

### Definition of control arm

Men randomised to the control arm undergoes systematic biopsies as defined below.

### Technology
#### Cut-offs for performing the STHLM3 test

The STHLM3 test will be performed for men with a PSA ≥1.5 ng/mL

#### Cut-offs for entering randomisation

Participants with PSA ≥3.0 ng/mL or STHLM3-test ≥11% risk of Gleason Score ≥7 cancer will be randomised and offered to undergo either MRI or systematic biopsies (see Recruitment and process description section).

### MRI technology

#### Location and MRI equipment
► Capio St Görans Hospital: General Electric, Architect, 3T.
► Globen Unilabs Healthcare: Siemens Magnetom Aera 1.5T.

#### Patient preparations
► Refraining from sexual activity with ejaculation 3 days prior to examination.
► Fasting patient for 6 hours.
► Minimal preparation enema prior to examination.
► Antispasmodic agent (Glucagon) just before the examination.

#### MRI Protocol

A short (14 min) MRI protocol will be used. A detailed description is available. Briefly, the protocol includes the following: T2w images axial, sagittal, coronal; diffusion-weighted imaging b0 and b1000 with apparent diffusion coefficient (ADC) and a synthetic b1500 limited to the prostate location; endorectal coil will not be used.

#### MRI Interpretation

MRI interpretation is centralised to Capio St Görans hospital and is performed according to PIRAD V.2.0 for examinations without adequate perfusion studies. Dr Fredrik Jäderling is responsible for MRI interpretation. Dr Jäderling or one to two other experienced radiologists at his department perform all MRI interpretations.

PI-RADS V.2 ('Assessment without adequate dynamic contrast enhanced imaging') will be used, with a 1–5 grade scale of suspicious lesions (1=clinically significant cancer is highly unlikely to be present, 5=clinically significant cancer is highly likely to be present).

During the study period, participating radiologist will have access to updated histology results of fusion biopsies to be able to adjust their MRI reading according to tumour detection rates for different PIRAD scores as defined above.

### Fusion biopsy technology

#### Brand/models

BK Medical (BK Ultrasound; www.bkultrasound.com/bk-medical/fusion)

The BK Medical fusion system is the only fusion device compatible with BK Medicals ultrasound devices, used by the urology departments participating in the study. The system represents a second-generation ultrasound system with integrated MRI Fusion. MRI data are imported through HIPAA-compliant PACS connection with the local radiology department.

#### Definition of targeted biopsies

Using MRI data with pre-marked borders of the prostate and tumour, fusion of MRI images and ultrasound images are performed bedside. Using local anaesthetics and antibiotic prophylaxis, lesions are taken according to the schedule below. Targeted biopsies are always combined with systematic biopsies.

#### Biopsy procedure for targeted biopsies

PI-RADS ≥3: Three to four targeted biopsies on marked lesions+systematic biopsies.

Large diffuse lesions or poor image quality: Systematic biopsies including lesion.

No PI-RADS ≥3, diffuse lesions and at least acceptable image quality: No biopsies are performed.

In larger lesions in PI-RADS categories 3 and 5, areas within the lesion with the lowest ADC value ('Target-within-target') will be targeted with the first biopsy taken from the lesion, to evaluate the additional value regarding tumour staging.

#### Definition of systematic biopsies

Ten to twelve systematic biopsies are taken from the peripheral zone as previously described in STLHLM3 and the national guidelines. Extra biopsies are allowed from additional sites visible on ultrasound or according to palpatory findings. In summary, systematic biopsies are performed in the peripheral zone as four lateral and para-median biopsies on the left and right sides, in the base and mid parts of the gland. In the apical third of the gland, one lateral left and right biopsy is performed.

### Pathology

Pathology is centralised to Unilabs/Capio St Görans hospital. Dr Axel Glaessgen is responsible for the integrity of analyses of pathological specimen. Two to three uropathologists at Dr Glaessgens department assess all pathological specimen with intermittent cross-validation between them. Pathology preparation and reporting follow ISUP 2014 guidelines.

The pathology preparation is done by Unilabs as part of the normal clinical routine. Biopsy specimens are analysed according to local practice.

Localisation of biopsies in the prostate are described using Swedish National Guideline nomenclature (A1-4; B1-4; C1-4; anterior/posterior). Gleason Score, mm cancer and % Gleason 4 are reported on each needle specimen.

Pathologist notes result in the usual way in the laboratory system. The result of the pathological analysis is submitted in accordance with existing clinical routines to the referring urologist. A copy of the result is delivered to the study administration.

### Outcomes

There are three co-primary endpoints in this trial: (1) number of diagnosed ISUP grade group ≥2 cancers; (2) number of diagnosed ISUP grade group 1 cancers and (3) number of performed biopsies.

### Follow-up

The main study outcomes are assessed after prostate biopsy procedures. Additional participant data will be secured in the following circumstances:

#### No suspicious lesion on MRI

Men in the experimental arm without suspicious lesions on MRI will be informed and recommended follow-up by the responsible, local urologist. After additional ethical application, the co-investigators may initiate retrospective follow-up of these participants.

#### Men with diagnosed prostate cancer

Participants with prostate cancer diagnosed on biopsy within the study will be followed up after the biopsy to secure data on the following: treatment modality (active surveillance, surgery, radiation); treatment lead-time and site; and pathological report after surgery (positive margins, T-stage, etc). Data will be assessed through medical records intermittently.

### Serious adverse events

A study nurse will monitor serious adverse events after the prostate biopsy procedures. To ensure this, the study nurse will check medical journals for hospitalisation within 1 week after the biopsy procedure in the journal systems *Take Care and Cosmic* (covering all hospitals in the Stockholm region). This will be initiated as individual biopsy results are registered at the study administration. Results will be provided to the Data Safety and Monitoring Board (DSMB).

### Participant timeline

Figure 2 illustrates the approximate timeline for men participating in the STHLM3MRI Main Study.

### Sample size

STHLM3-MR/Fusion phase II will invite 25 000 men and aim to include 10 000 participants. We anticipate to perform 1039 biopsy procedures altogether. Inclusion will continue until complete data are available on 415 men in the control arm (SBx) and 623 men in the experimental arm (MR-TBx-SBx).

### Basic data and assumptions used in the sample size calculations

We used data from the STHLM3 trial for sample size calculations.[7] In these data, 18% of men with PSA ≥3 had a clinically significant prostate cancer when biopsied with SBx. We further noted that rTPR=1.45 for clinically significant prostate cancer comparing MRI+TBx with SBx based on the results from the PRECISION randomised trial.[20] However, for sample size calculations, we will use rTPR=1.25 for MRI+TBx vs SBx as a more conservative estimate. We set the non-inferiority delta to 4 percentage points for demonstrating non-inferiority with respect to sensitivity of clinically significant prostate cancer. We set the alpha to 5%.

### Primary contrast

Simulating 1000 trials (by bootstrapping from the STHLM3 data) under the assumptions outlined in the preceding section, 303 men need to be biopsied in the SBx arm based on PSA ≥3 to have 80% power to demonstrate non-inferior sensitivity of S3M+MRI+TBx

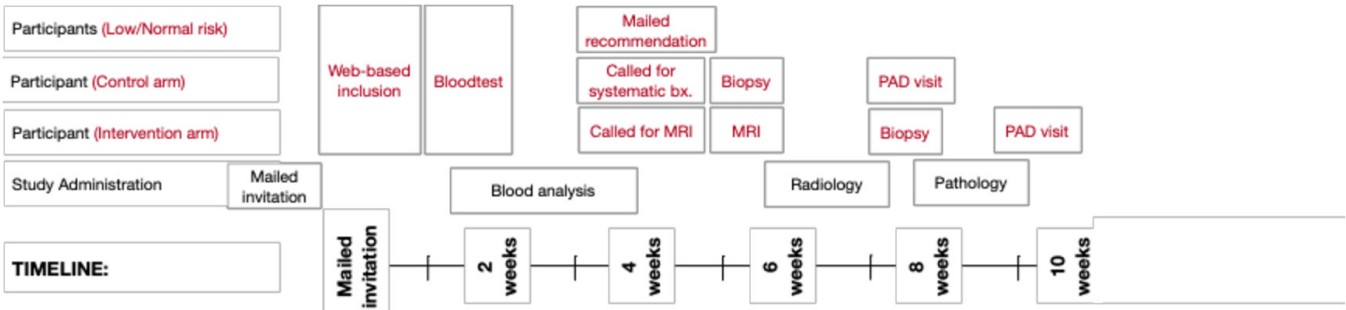

**Figure 2** Timeline overview for study participants in the STHLM3MRI Main Study.

compared with PSA+SBx. This means that at least 415 men need to be biopsied in the SBx arm (since some men are not randomised based on PSA ≥3 but on S3M≥11%) and, consequently, 623 to the MRI arm (because of the 2:3 randomisation). The total number of men undergoing workup according to protocol (SBx in the no MRI arm and MRI and TBx if PI-RADS ≥3 in the MRI arm) is thus 1038. Assuming 20% dropout, 1300 men need to be randomised. These numbers give 80% power to detect a modest 17% reduction in biopsies between the two strategies.

### Recruitment and process description

The STHLM3-MR phase II will use existing solutions developed and optimised in the previous studies STHLM3 and STHLM3-MR phase I, where all major components of the process have been tested. First, participants will follow the paired design study process where inclusion, blood test and delivery of recommendation letter are performed. Men with increased risk of high-grade prostate cancer then enter the randomised study process, where extended workup including biopsies are performed.

### Data collection, management and analysis

#### Data collection

Primary data sources are as follows:

1. 1. Clinical variables collected from laboratory referral.
2. 2. Biopsy referrals and reports.
3. 3. Pathology reports.
4. 4. MRI reports.
5. 5. Blood concentrations of kallikreins, MSMB, MIC1 SNPs.

Collection of (1) to (4) is performed by study nurses (C Cavalli-Björkman) on a weekly basis from participating urology sites, participating radiologists. For (5), this is digitally transferred from Karolinska University Laboratory.

#### Data management

Data are collected, entered, coded and stored at Department of Medical Epidemiology and Biostatistics, Karolinska Institutet. Data are entered by a study nurse using predefined database sheets developed in STHLM3MRI phase I. This is blinded from study co-investigators and data are stored at the department under supervision by the SDA (Astrid Björklund). Any extraction of study data is performed by the SDA after approval of PI Tobias Nordström.

#### Data analysis

Analysis of data is described in the Statistical Analysis Plan.

#### Auditing and monitoring

A DSMB is assembled and consists of Dr Hans Garmo (Statistician), Professor Ola Bratt (Urology) and Professor Holmberg (Urology/Study Design). The DSMB audits protocol and process descriptions and one interim data extraction performed by the SDA after 10% (100 men)

have completed the control or experimental arms. The co-investigators are blinded to the interim data and analysis results. The work of the DSMB is regulated in the DSMB Charter.

### Patient and public involvement

The research question and outcome measures were designed to improve prostate cancer diagnostics. This includes optimising prostate biopsies and decreasing overdetection, both associated with morbidity. Patient organisations were informed on the results from the STHLM3MRI phase I study. Patients were not involved in recruitment of the study. Results will be disseminated to participants through common and scientific channels.

### Consent

Participant consent is secured when the participant is included to the study at (www.kliniskastudier.se). This includes secure identification using Mobilt BankID. Additional approval on use of biological specimen data is collected on the biopsy referral.

### Confidentiality

Study data are collected and stored at Department of Medical Epidemiology and Biostatistics, Karolinska Institutet using secure Oracle servers. All data extractions are made by database administrator and are anonymised (personal id number is removed) before dissemination to researchers.

### Dissemination

Analyses results on the posed aims will be submitted for peer-reviewed publication and submitted for presentation at scientific congress. Communication of the results will be made to patient organisations (Prostatacancerförbundet) and non-scientific channels. No use of professional writers is planned.

The study protocol is made publicly available through (clinicaltrials.gov).

**Author affiliations**
[1]Department of Medical Epidemiology and Biostatistics, Karolinska Institutet, Solna, Sweden
[2]Department of Clinical Sciences at Danderyd Hospital, Karolinska Institutet, Solna, Sweden
[3]Department of Molecular Medicine and Surgery, Karolinska Institutet, Stockholm, Sweden
[4]Department of Diagnostic Radiology, Karolinska University Hospital, Stockholm, Sweden
[5]Patient area Pelvic Cancer, Theme Cance Karolinska University Hospital Solna, Stockholm, Sweden

**Acknowledgements** We thank participants, study organisers, participating researchers and clinicians and patient advisers for their contributions to the STHLM3MRI project.

**Contributors** TN was the Principal investigator. TN, HG, ME, SC and MA designed the study. ME and TN interpreted preliminary data. FJ designed MRI protocols and collected data.

**Funding** Funding was provided by the Swedish Cancer Society (Cancerfonden), the Swedish Research Council (Vetenskapsrådet), Swedish Research Council for Health Working Life and Welfare (FORTE), The Strategic Research Programme on Cancer (StratCan), Karolinska Institutet, Swedish e-Science Research Center (SeRC) and

Stockholm City Council (SLL). The STHLM3 study is a part of the Linnaeus Center CRISP 'Predication and prevention of breast and prostate cancer' funded by the Swedish Research Council.

**Competing interests** HG has five prostate cancer diagnostic related patents pending, has patent applications licensed to Thermo Fisher Scientific and might receive royalties from sales related to these patents. ME is named on four of these five patent applications. Karolinska Institutet collaborates with Thermo Fisher Scientific in developing the technology for the Stockholm3 test.

**Patient consent for publication** Not required.

**Ethics approval** The study has approval from the Regional Ethical Review Board in Stockholm (2017-1280/31).

**Provenance and peer review** Not commissioned; externally peer reviewed.

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
