## [Reviewer comments · BMJ Open]

ARTICLE DETAILS

TITLE (PROVISIONAL)	Does a novel diagnostic pathway including blood-based risk-prediction and MRI-targeted biopsies outperform prostate cancer screening using prostate-specific antigen and systematic prostate biopsies? – Protocol of the randomized study STHLM3MRI.
AUTHORS	Nordstrom, Tobias; Jäderling, Fredrik; Carlsson, Stefan; Aly, Markus; Grönberg, H; Eklund, Martin

VERSION 1 - REVIEW

REVIEWER	Mark EMBERTON UCL, United Kingdom Research on new screening methods is a research interest. I know many of the authors personally.
REVIEW RETURNED	04-Jan-2019

GENERAL COMMENTS	This is a most interesting and timely design. The manuscript is very clear. I feel it could be improved by addressing two points. 1. Is it reasonable to randomise men to PSA-TRUS biopsy as we know from two very large studies that this approach to screening fails to identify men at risk of premature death?2. The justification for the non-inferior analysis could do with some further comment. We need something distinctly better than PSA-TRUS biopsy. Equivalence would not be sufficient, in many peoples' view
--

REVIEWER	Caroline M Moore University College London United Kingdom
REVIEW RETURNED	02-Feb-2019

GENERAL COMMENTS	This paper describes an important screening study for prostate cancer which aims to assess a smarter screening strategy using a combination of PSA and STHLM3 to determine who needs further procedures (MRI or biopsy). It is clearly written. Minor comments:
--

	P15, line 26 Typo - should be investigator p16 - references need clarifying/given numerical reference The author V Kasivisvanathan has been shortened to Kasi - full name should be used.
--	---

VERSION 1 – AUTHOR RESPONSE

Reviewer: 1

This is a most interesting and timely design. The manuscript is very clear.

Authors comment: Thank you!

I feel it could be improved by addressing two points.

1. Is it reasonable to randomise men to PSA-TRUS biopsy as we know from two very large studies that this approach to screening fails to identify men at risk of premature death?

Authors comment: We acknowledge this point. Nonetheless, the diagnostic strategy of using PSA (>3ng/ml) for identifying men at increased risk and then performing systematic biopsies still is the only strategy that have been shown to decrease the risk of metastatic disease and prostate cancer death (Ref ERSPC trial, including the Gothenburg study). Further, while imaging-based targeted biopsies are evolving, PSA-based strategies combined with systematic biopsies are still standard of care in many clinical contexts.

Thus, after careful discussions within the study group, we have decided to use PSA+systematic biopsies as our comparator for the primary analyses. However, the study design enables also other comparisons (e.g. PSA+systematic biopsies vs PSA+targeted biopsies) as indicated in 5.1.2 Additional hypotheses.

These arguments will be published alongside the manuscript, and in addition, we have inserted a line clarifying this in the manuscript (4.3):

“Since screening using PSA and systematic prostate biopsies have been shown to decrease prostate cancer mortality, it is reasonable to use this strategy as comparator for novel diagnostic strategies.”

2. The justification for the non-inferior analysis could do with some further comment. We need something distinctly better than PSA-TRUS biopsy. Equivalence would not be sufficient, in many peoples' view

Authors comment: Yes, we agree that this can be clarified. While there is a mortality benefit shown in the ERSPC trial, we argue that non-inferior sensitivity to detect significant disease is a relevant baseline. However, for the novel strategy to outperform e.g. PSA+systematic biopsies, we need to show

superior specificity, e.g less performed biopsies and less detected low-grade cancers. Thus, we use three co-primary endpoints (5.4.5 Endpoints) – nr of detected ISUP \geq 2 (non-inferior sensitivity), nr of detected ISUP 1 (superior specificity), nr of biopsies (superior specificity).

We have adjusted the text in the Primary Hypotheses to clarify this:

“A diagnostic pathway using the Stockholm3 test to select men for further workup using MRI followed by targeted biopsies and systematic biopsies (S3M-MR-TBx/SBx) has non-inferior sensitivity for detecting clinically significant cancer (ISUP grade group \geq 2) and shows superior specificity (reduction in number of performed biopsy procedures and detected ISUP 1 tumors) compared to a diagnostic pathway using systematic biopsies in men with PSA \geq 3 ng/ml (PSA-SBx).”

Reviewer: 2

This paper describes an important screening study for prostate cancer which aims to assess a smarter screening strategy using a combination of PSA and STHLM3 to determine who needs further procedures (MRI or biopsy).

It is clearly written.

Authors comment: Thank you!

Minor comments:

#1: P15, line 26 Typo - should be investigator

#2: p16 - references need clarifying/given numerical reference

#3: The author V Kasivisvanathan has been shortened to Kasi - full name should be used.

Authors comment: Adjustments have been made.

VERSION 2 – REVIEW

REVIEWER	Mark EMBERTON UCL, United Kingdom I am an active researcher in this area, cited by the authors most of whom I know well.
REVIEW RETURNED	29-Mar-2019

GENERAL COMMENTS	This a well written and precise account of a study that wishes to compare a novel risk-stratification strategy with PSA and TRUS biopsy. The main methodological issue with the study relates to the choice of control. We have a number of studies that show
---

	superiority of an MRI directed biopsy compared to the standard of care PSA TRUS biopsy. The world moves on quickly these days and NICE now recommend an MRI prior to biopsy in all men. This means that in the UK and by the time this study gets reported PSA TRUS biopsy will no longer be the standard of care. Is it not the case that the modern, timely and preferred design would now be a randomisation between STHLM3 and MRI targeted biopsy versus MRI targeted biopsy alone?
--	---

VERSION 2 – AUTHOR RESPONSE

Reviewer(s)' Comments to Author:

This a well written and precise account of a study that wishes to compare a novel risk-stratification strategy with PSA and TRUS biopsy. The main methodological issue with the study relates to the choice of control. We have a number of studies that show superiority of an MRI directed biopsy compared to the standard of care PSA TRUS biopsy.

The world moves on quickly these days and NICE now recommend an MRI prior to biopsy in all men. This means that in the UK and by the time this study gets reported PSA TRUS biopsy will no longer be the standard of care.

Is it not the case that the modern, timely and preferred design would now be a randomisation between STHLM3 and MRI targeted biopsy versus MRI targeted biopsy alone?

Author response: Thank you! We really appreciate this insightful comment. During the design phase, we discussed study endpoints carefully and present a summary of our argumentation here.

-We agree that detection of prostate cancer is rapidly evolving, including the introduction of pre-biopsy MRI and targeted biopsies in guidelines and clinical practice. However, we also note that systematic biopsies without MRI still is widely employed in a wide variety of international clinical settings. We believe this will be the case for some time, especially in the light of limited MRI resources in many countries. Therefore, systematic biopsies are still relevant to use as a comparator.

- Level 1 evidence of a mortality benefit from early detection of prostate cancer is still only presented for a strategy using PSA and systematic biopsies (ERSPC). Therefore, we argue that PSA + systematic biopsies still is the first-hand choice for control arm to use in any trial on novel diagnostic strategies.

-Further, contrasting to the excellent PRECISION trial (NEJM 2018) where men with a clinical suspicion of prostate cancer were included, there is still a lack of comparisons between MRI/Fusion

biopsies and systematic biopsies using randomized screen-by-invitation designs. We believe that screening populations might differ from clinical cohorts in several aspects (disease prevalence, PI-RADS distribution, MRI lesion size etc). Thus, we argue that it is important to prove the performance of MRI/targeted biopsies vs systematic biopsies also in a well-controlled screening context.

-Nonetheless, our study design permits the suggested comparison within the experimental arm using a paired design where the performance of the strategy PSA+MR/targeted biopsies is compared to the strategy Stockholm3+MR/targeted biopsies. This is stated in Additional Hypotheses 4 (row 252). Due to text limitations, the corresponding Aim is only referred to as “Additional aims corresponding to hypotheses 2-8 above will be assessed.” (row 274). From a methodological view-point, this analysis is similar to the one used in the original STHLM3 study (Lancet Onc 2015). Thus, we plan to present the analysis as a separate analysis. While the study was not primarily powered for this comparison, we are currently re-assessing the power analysis for this analysis.